# Proposed Mobility Assessments with Simultaneous Full-Body Inertial Measurement Units and Optical Motion Capture in Healthy Adults and Neurological Patients for Future Validation Studies: Study Protocol

**DOI:** 10.3390/s21175833

**Published:** 2021-08-30

**Authors:** Elke Warmerdam, Robbin Romijnders, Johanna Geritz, Morad Elshehabi, Corina Maetzler, Jan Carl Otto, Maren Reimer, Klarissa Stuerner, Ralf Baron, Steffen Paschen, Thorben Beyer, Denise Dopcke, Tobias Eiken, Hendrik Ortmann, Falko Peters, Felix von der Recke, Moritz Riesen, Gothia Rohwedder, Anna Schaade, Maike Schumacher, Anton Sondermann, Walter Maetzler, Clint Hansen

**Affiliations:** Department of Neurology, Kiel University, Arnold-Heller-Str. 3, 24105 Kiel, Germany; r.romijnders@neurologie.uni-kiel.de (R.R.); j.geritz@neurologie.uni-kiel.de (J.G.); m.elshehabi@neurologie.uni-kiel.de (M.E.); c.maetzler@neurologie.uni-kiel.de (C.M.); jc.otto@gmx.net (J.C.O.); maren.reimer@uksh.de (M.R.); klarissa.stuerner@uksh.de (K.S.); r.baron@neurologie.uni-kiel.de (R.B.); s.paschen@neurologie.uni-kiel.de (S.P.); thorben.beyer@uksh.de (T.B.); denise.dopcke@uksh.de (D.D.); tobias.eiken@uksh.de (T.E.); janhendrik.ortmann@uksh.de (H.O.); falko300@t-online.de (F.P.); felix.vdr@gmx.de (F.v.d.R.); moritz.riesen@uksh.de (M.R.); gothia.rohwedder@uksh.de (G.R.); anna.schaade@uksh.de (A.S.); maike.schumacher@uksh.de (M.S.); antonsondermann@yahoo.de (A.S.); w.maetzler@neurologie.uni-kiel.de (W.M.); c.hansen@neurologie.uni-kiel.de (C.H.)

**Keywords:** balance, chronic low back pain, gait, movement analysis, multiple sclerosis, Parkinson’s disease, stroke, wearable sensors

## Abstract

Healthy adults and neurological patients show unique mobility patterns over the course of their lifespan and disease. Quantifying these mobility patterns could support diagnosing, tracking disease progression and measuring response to treatment. This quantification can be done with wearable technology, such as inertial measurement units (IMUs). Before IMUs can be used to quantify mobility, algorithms need to be developed and validated with age and disease-specific datasets. This study proposes a protocol for a dataset that can be used to develop and validate IMU-based mobility algorithms for healthy adults (18–60 years), healthy older adults (>60 years), and patients with Parkinson’s disease, multiple sclerosis, a symptomatic stroke and chronic low back pain. All participants will be measured simultaneously with IMUs and a 3D optical motion capture system while performing standardized mobility tasks and non-standardized activities of daily living. Specific clinical scales and questionnaires will be collected. This study aims at building the largest dataset for the development and validation of IMU-based mobility algorithms for healthy adults and neurological patients. It is anticipated to provide this dataset for further research use and collaboration, with the ultimate goal to bring IMU-based mobility algorithms as quickly as possible into clinical trials and clinical routine.

## 1. Introduction

Healthy adults, as well as patients with neurological diseases, such as Parkinson’s disease (PD), stroke and multiple sclerosis (MS), show unique mobility patterns over the course of their life span and disease. These unique mobility patterns can be used for diagnosis [1,2], tracking disease progression [3], measuring efficacy of treatment [4] and detecting side effects of chronic medication intake [5]. In clinical routine, mobility patterns are generally evaluated by healthcare professionals during a clinical or in-praxis examination. Objective evaluation methods can provide additional and potentially more ecologically valid measures. Wearable technology, more specifically inertial measurement units (IMUs) are highly suited for objective movement analysis and can even be used to analyse mobility patterns outside the clinic and praxis, i.e., the usual environment [6,7].

Currently, results of such mobility analyses differ substantially between different IMU devices [8,9]. This is most likely due to multiple reasons, including lack of standardization of IMU position on the body and lack of (disease-) specific and thorough validation of the algorithms used to extract and analyse raw data. Thus, before these IMUs are used in the natural environment of the healthy adults and patients, clear information about the best position of the IMUs to calculate mobility-related parameters should be gathered and a thorough and specific validation of the used algorithms must be performed [6,10,11].

The accuracy of algorithms for the analysis of IMU-derived data is dependent on laborious validation studies, which cannot be performed in every laboratory and specifically for every single research question. In such validation studies, these IMU-derived algorithms need to be compared to data extracted from reference tools for the assessment of mobility, such as 3D optical motion capture systems. As mobility patterns differ across lifespan and between different neurological diseases, this validation must be performed in different age groups and in disease-specific datasets. To our best knowledge, there is currently no representative dataset available that allows for such validation by providing multiple IMU positions in a variety of neurological diseases. We propose here a study protocol to build a full-body mobility dataset of healthy young and older participants and neurological patients, including PD, MS, stroke and chronic low back pain (CLBP). All participants will be measured simultaneously with 15 IMUs and 47 reflective markers that are tracked with a 3D optical motion capture system. The assessment will include standardized mobility tasks as well as non-standardized activities of daily living. Specific clinical scales will be provided as anchors. The aim of the study is to build a dataset for the research community that can be used to develop and validate IMU-based mobility algorithms for healthy adults and neurological patients.

## 2. Materials and Methods

### 2.1. Ethics

This study is approved by the ethical committee of the Medical Faculty of Kiel University (D438/18) and is in accordance with the principles of the Declaration of Helsinki. All participants will receive written and oral information about the measurements. The participants will have to provide written informed consent before the start of the measurements. The study is registered in the German Clinical Trials Register (DRKS00022998).

### 2.2. Participants

This study will include healthy adults (18–60 years), healthy older adults (>60 years), and patients with PD (according to the UK Brain Bank Criteria [12]), MS (according to the McDonalds criteria [13]), patients with a recent (<4 weeks) symptomatic stroke and patients with CLBP, whose patients characteristics are described elsewhere [14]. Healthy adults will be recruited via flyers that will be placed in public facilities. Neurological patients will be recruited from the neurology wards and outpatient clinics of the University Hospital Schleswig-Holstein (UKSH), Campus Kiel, Germany. Inclusion criteria are 18 years and older, and the ability to walk independently without walking aid. Exclusion criteria are a Montreal Cognitive Assessment score <15 and other movement disorders that affect mobility performance, as judged by the assessor.

### 2.3. Clinical and Demographic Data

Demographic data, including age, gender, weight, height, foot size, handedness, will be recorded. Furthermore, comorbidities of all participants will be assessed with the Charlson Comorbidity Index [15]. The cognitive function will be assessed with the Montreal Cognitive Assessment [16]. Generic health status will be assessed with the EQ-5D-5L [17]. Activities of daily-living will be assessed with the Lawton Instrumental Activities of Daily Living Scale [18] and the German Funktionsfragenbogen Hannover [19]. Sarcopenia will be assessed with the SARC-F [20]. Pain will be assessed with the Visual Analogue Scale [21]. Vibratory sensation will be assessed with a tuning fork (Rydel-Seiffer) [22]. Fatigue will be assessed with the Fatigue Severity Scale [23]. The perceived self-efficacy will be assessed with the General Self-Efficacy Scale [24]. The motor function of all participants will be assessed with the motor part of the Movement Disorders Society Sponsored Revision of the Unified Parkinson’s Disease Rating Scale (MDS-UPDRS) [3].

### 2.4. Disease Specific Scales

From all the patient with a neurological disorder, the diagnosis, disease duration as well as the medication (type, dose and frequency) that the patients take will be collected from the medical record. Additionally, for the PD patients the Hoehn and Yahr stage [25], for the MS patients the Expanded Disability Status Scale [26] and for the stroke patients the NIH Stroke Scale will be assessed [27].

### 2.5. Equipment

Participants will be measured with IMUs (Noraxon USA Inc., myoMOTION, Scottsdale, AZ, USA), containing a triaxial accelerometer (+/− 16 g), triaxial gyroscope (+/− 2000 degrees/sec) and triaxial magnetometer (+/− 1.9 Gauss). A total of 15 IMUs will be attached to different body segments (Figure 1a). IMUs are therefore fixed to the following body segments: head, sternum, upper arms, fore arms, pelvis, thighs, shanks (proximal), ankles and feet. The IMUs will be secured with elastic bands with a special hold for the IMU attached to it. In case the participant has pockets in the shorts, a 16th IMU will be placed in the pocket. The data from this 16th IMU could be used to develop and validate algorithms for smartphones that are commonly worn in the pocket. The IMU data will be collected with a sample frequency of 200 Hz.

As reference, a twelve-camera optical motion capture system (Qualisys AB, Göteborg, Sweden) will be used to record full-body movements with 200 Hz. A total of 47 reflective markers (19 mm) will be adhered to the body (Figure 1b) for all movement assessments. A minimum of three markers can be found on the following body segments: head, sternum, upper arms, fore arms, hands, lower back, thighs, shanks and feet. During static calibration trials, 8 additional reflective markers (19 mm) will be placed on the body (elbows, knees and ankles) to be able to estimate joint positions (the exact positions of all the reflective markers are described in the Appendix A). The IMU data and the optical data will be synchronized with help of a TTL signal.

Two reflex light barriers (Telemecanique, photo-electronic sensor XULM06031, Rueil-Malmaison, France) standing 5 m apart will be used to measure the preferred over ground gait speed.

The over ground walking will be performed on a walkway with a width of 1 m. The start and end of the 5 m during which steady state gait is recorded will be marked by cones with reflective markers (30 mm) on top of them. For the assessment of longer gait bouts a treadmill (Woodway, Waukesha, WI, USA) of 2.10 by 0.70 m with a split belt option will be used. Dual-task assessments during over ground walking will be performed on a smartphone with a screen size of 4.5 inch (One Touch Pop 2, Alcatel, Hong Kong, China). A simple reaction time test and a numerical Stroop test will be used as dual-task (developed with https://www.neurobs.com/menu_presentation/menu_features/mobile, accessed on 10 February 2021).

The whole assessment of each participant will be videotaped by two cameras (GoPro Inc., Hero Session, San Mateo, CA, USA). The videos will be synchronized with the IMUs and optical data with help of a synchronization light that turned on and off at the start and end of each measurement.

### 2.6. Protocol

During the assessment of patients, there will always be a staff member standing close to the participant to support the patient in case of a loss of balance. Patients with PD will be asked to perform the whole protocol part twice, both on and off dopaminergic medication. An overview of the protocol is given in Figure 2.

At the start, the preferred over ground speed will be measured with reflex light barriers. Participants will start walking about 2 m before the first light barrier and will stop walking about 2 m after the second light barrier. The average gait speed of five trials will be calculated and used as walking speed on the treadmill.

All trials listed below will be recorded with IMUs and the optical motion capture system.

Each assessment starts with a calibration trial where participants stand in a neutral pose (feet at hip width and arms hanging along the body). This trial will be repeated every time an IMU or marker is displaced. This calibration trial can be used to define the anatomical reference frame with respect to the technical reference frame for both the IMU and the optical motion capture system [28,29,30]. Next the MDS-UPDRS part III will be assessed. These trials will always be performed in this fixed order at the beginning of the measurement. Hereafter, the standardized and non-standardized mobility assessments will be performed in randomized order.

#### 2.6.1. Standardized Mobility Assessments

Treadmill walking. All participants will wear a safety harness that is suspended from the ceiling while walking at the treadmill. At the start of the treadmill trial, the speed of the treadmill will be gradually increased to a speed that is comfortable for the participant. The participant will walk for 60 s at this speed. Thereafter, the speed of the treadmill will be gradually adapted to the preferred over ground walking speed which is measured at the start of the protocol. The participant will walk again 60 s at this speed. A subset of the healthy young adults will participate in a split-belt protocol which is described in the Appendix AShort physical performance battery (SPPB)○Side-by-side stand (“Please stand with your feet together for 10 s, try not to move your feet”)○Semi-tandem stand (“Please stand with the heel of one foot touching the big toe of the other foot for 10 s, you can put either foot in front, try not to move your feet”)○Tandem stand (“Please stand with the heel of one foot in front while touching the toes of your other foot, you can put either foot in front, try not to move your feet”)○m gait (“Please stand with the toes of both feet on the starting line and walk over to the end of the walkway at your normal gait speed”)○m gait (“Please stand again with the toes of both feet on the starting line and walk over to the end of the walkway at your normal gait speed”)○Repeated chair Stand (“Please stand up straight five times in a row as fast as possible without using your arms”)Timed up and go (“Please stand up from the chair, walk at preferred speed towards the cone, turn around it in the direction of your preference, walk back and sit down”)Five time sit to stand test (“Please stand up straight five times in a row at your preferred speed without using your arms if possible”)“Choreography”: a series of movements related to the flexibility of the lower back (see Appendix A). The choreography contains flexion, extension and rotational movements of the back, as well as a combination of those movements (“Please perform the movements that are shown one by one on the pictures”)

The following standardized walking assessments will take place on the 5 m walkway (Figure 3). All participants will be asked to start two steps before the start of the walkway and stop walking two steps after the end of the walkway.

Straight walking○Slow speed (“Please walk half of your normal walking speed”; Figure 3a)○Preferred speed (“Please walk at your normal walking speed”)○Fast speed (“Please walk as fast as possible, without running or falling”)Sideways walking (“Please walk sideways, do not cross your legs during this walk”)Backwards walking (“Please walk backwards at a speed that is comfortable for you”)Obstacles: an obstacle with a height of 10 cm, and one with a height of 20 cm will be placed at the three meter point with reflective markers on the top of each side (Figure 3b), and a forward walk will be performed once for each obstacle (“Please walk at your normal walking speed and step over the obstacle”)Slalom: cones will be placed every meter in the middle of the walkway. Each cone will have a reflective marker on top (“Please walk at your normal speed around the cones, do not step over them”; Figure 3d)Single and dual-tasking: It is know that the complexity of the dual-task influences the dual-task costs [31,32], therefore two tasks with different complexity will be performed. The first task will be a simple reaction time test where participants will have to tap on the screen as fast as possible after a black square appears on the screen. There are six time intervals ranging from 1000 to 2000 ms (increased in steps of 200 ms), which determines the time it will take for the black square to appear on the screen. Each time interval occurs four times and the order of the 24 options is randomized. The reaction time will be recorded. A more complex reaction time test that is more often used to measure the dual-task performance is the Stroop test [33,34,35]. The Stroop test also measures the cognitive inhibition [35,36]. In this study a numerical Stroop test will be performed. During this test two numbers will appear on the screen and the participants have to tap on the number that is highest in value. Within this test there are three conditions; (1) Neutral, the font size of both numbers is equal; (2) Congruent, the number highest in value has a larger font size; (3) Incongruent, the number highest in value has a smaller font size (Figure 4). In total 24 responses will be required, eight of each condition. The order in which the 24 options occur in the test is randomized. The reaction time as well as the accuracy will be recorded.○Simple reaction time task on a smartphone while standing (“Please tap on the screen as fast as possible after a black square appears on the screen”)○Numerical Stroop task on a smartphone while standing (“On the screen will appear each time two numbers, please tap on the largest number in value, not the largest number in size”)○Walking up and down the 5 m walkway for 30 s, turning direction was not instructed (“Please walk up and down the walkway at your normal speed and stay within the area marked by the cones”; Figure 3d)○Walking up and down the 5 m walkway and performing the simple reaction time test on the smartphone (“Please perform the simple reaction time test again as instructed before and walk up and down the walkway at your normal speed at the same moment”)○Walking up and down the 5 m walkway and performing the Numerical Stroop test on the smartphone (“Please perform the numerical Stroop test again as instructed before and walk up and down the walkway at your normal speed at the same moment”)

#### 2.6.2. Non-Standardized Activities of Daily Living Assessment

The non-standardized mobility assessment consists of common daily activities that will be performed by the participants. The daily activities that will be performed are listed below. The order of the activities will not be fixed and will be decided by the researcher in the flow of this assessment:Setting a table (plates, cutlery, glasses)Eating and drinking (including opening a bottle and pouring a drink)Cleaning a tableLifting/replacing objects from different heightsIroning and folding a T-shirtTooth brushingMultiple chair risesSitting and reading out loudSitting and talkingOpening a cabinet and taking objects out of it

### 2.7. Database and Data Availability

The demographics and clinical data will be stored in a research electronic data capture (REDCap) database hosted at Kiel University [37]. This data will be shared upon reasonable request.

The IMU and optical data will be stored on a server of Kiel University that is only accessible by the research team. These data will also be made available online. The data of the first five healthy young adults and five older adults that have been measured already are available as “.mat” files (https://github.com/neurogeriatricskiel/Validation-dataset, accessed on 24 August 2021). More information about the data files can be found in the Appendix A. The videos that will be recorded will be stored on a separate server of the University Hospital of Kiel and will only be accessible to a small part of the research team. The videos will not be shared since they contain identifying information.

## 3. Discussion

This study will collect full-body mobility data from healthy young, older adults, and patients with PD, MS, stroke and CLBP. Each participant group will contain at least 20 participants with a maximum of 200 participants in total. All participants will be simultaneously measured with IMUs and optical motion capture. To our knowledge, this will be the first mobility dataset with full-body IMU and optical motion capture of healthy adults and multiple neurological patient groups of such size. The dataset can be used to develop and validate IMU-based algorithms for people with and without neurological diseases. With validated algorithms it will become possible to analyse mobility patterns both in the clinic and in the natural environment [6]. This objective information could help with diagnosing [1,2], tracking disease progression [3] and measuring the response to treatment [4,5].

Other studies with full-body IMU and optical motion capture included only young healthy participants [38,39,40]. Moreover, the participants performed a limited number of tasks that were not always mobility-related. Studies with full-body IMUs measuring either mobility-related tasks in older adults or symptoms in PD patients did not measure simultaneously with optical motion capture [41,42,43]. Other mobility related-studies that validated IMU-based algorithms against optical motion capture only measured the lower body simultaneously with both systems [44,45]. The upper body can however also provide relevant information regarding mobility [4,46].

The data that will be collected within this study will contain full-body IMU and optical motion capture data from a range of mobility-related tasks performed by both healthy participants and multiple neurological patient groups. Therefore, new and valuable information will be added to already existing datasets.

A large amount of standardized mobility assessments will be performed. There will be short (5 m) walking trials with different types of walking (straight, backwards, slalom, obstacle, sideways, dual-tasking). This will make it possible to test the accuracy of algorithms during straight walking and more complex walking assessments, which are likely to influence gait patterns [47,48,49]. To analyse the performance of algorithms during longer walks, there will be treadmill data collected. The split-belt treadmill walking data (speed reduction of 25% on one side [50]) can be used to analyse how well an algorithm deals with gait asymmetry. The SPPB, timed up and go and five chair rise test are well known assessment tools that are frequently performed in the clinic [51,52,53]. More information from these tests can be extracted by adding one or a few IMUs [54,55]. The non-standardized assessment part with activities of daily living can be used to develop and validate algorithms for the analysis of the performance in the natural environment of the patients. The different movements performed throughout all the assessments and the IMUs on different body parts make it also possible to define which IMU position is the most accurate to quantify a certain movement.

With the data from the different groups, disease-specific mobility patterns can be extracted and compared between diseases. These disease-specific mobility patterns could help to correctly diagnose patients [1,2]. It will also be possible to analyse how these mobility patterns change during the courses of the diseases, since the PD and MS groups will include patients with different disease stages [56]. All PD patients that consent in conducting assessments during ON and OFF dopaminergic medication states, will be measured in both conditions. This data will help assessing the effect of novel mobility algorithms and parameters to measure effect of treatment [4].

With the data from the different assessments it will be possible to analyse mobility in different circumstances. With the dual-task assessments it will, for example, be possible to measure how much the mobility deteriorates with an easy and a more complex dual-task. The walk with a low and a high obstacle will provide information about the obstacle negotiation performance, which could indicate whether the individual has an increased risk of falling [57,58]. Moreover, the clinical scores and questionnaires can be related to the mobility performance during the different assessments [59,60].

This study will have some limitations. The laboratory where the assessments will be performed is relatively small. Therefore, only 5 m of steady state walking can be captured on the over ground walkway and the distances covered during the non-standardized activities of daily living will also not exceed the 5 m. The measurements will last about three hours because of the many assessments that will be performed. It is possible that not every participant will be able to perform all assessments due to fatigue or loss of motivation, and that only a subset of the data can therefore be collected for those participants.

## 4. Conclusions

This study aims at building the largest currently available database for future development and validation of IMU-based mobility algorithms. It will include representative numbers of healthy adults over a large age range, as well as patients with diverse neurological diseases. The combined analysis of demographic and clinical data with full-body IMU and optical motion capture data should stimulate highly efficient research in this area, to eventually catalyse the implementation of accurate mobility parameters in clinical routine and assessment panels of clinical trials.

## Figures and Tables

**Figure 1 sensors-21-05833-f001:**
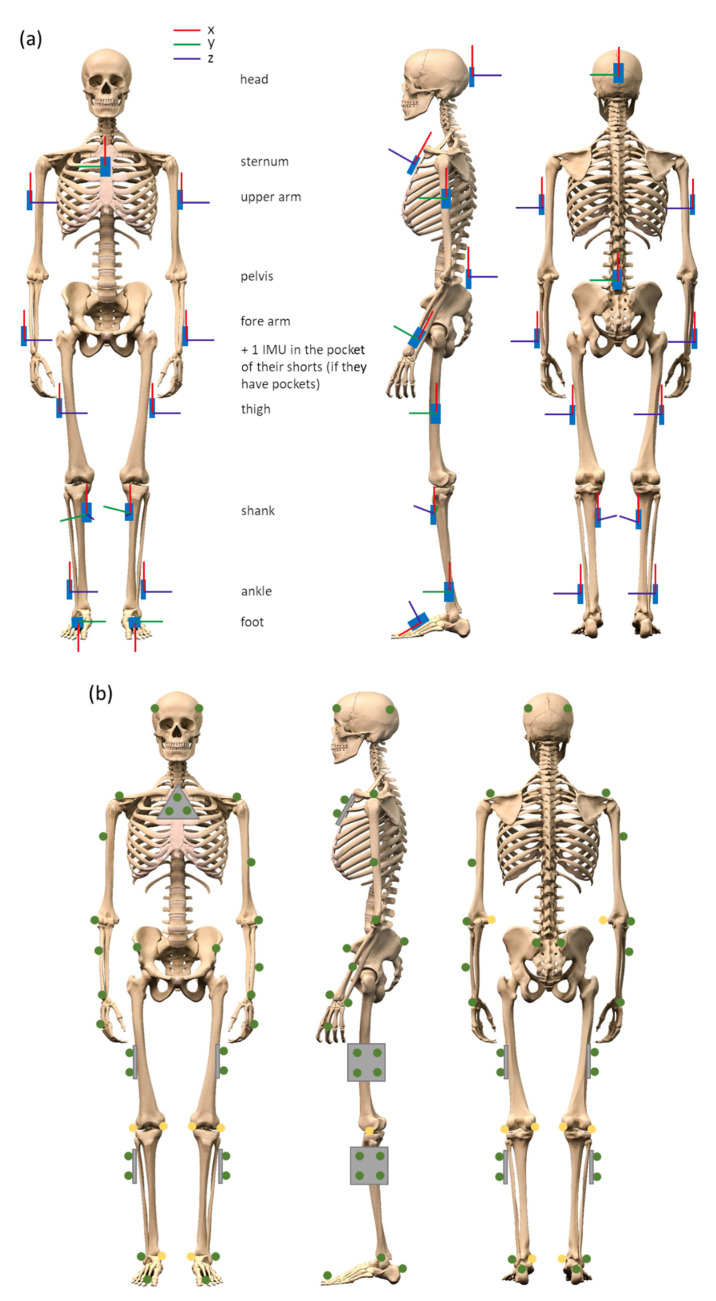
(**a**) Placement of inertial measurement units (IMUs) including the orientation. (**b**) Placement of the reflective markers measured by the optical motion capture system.

**Figure 2 sensors-21-05833-f002:**
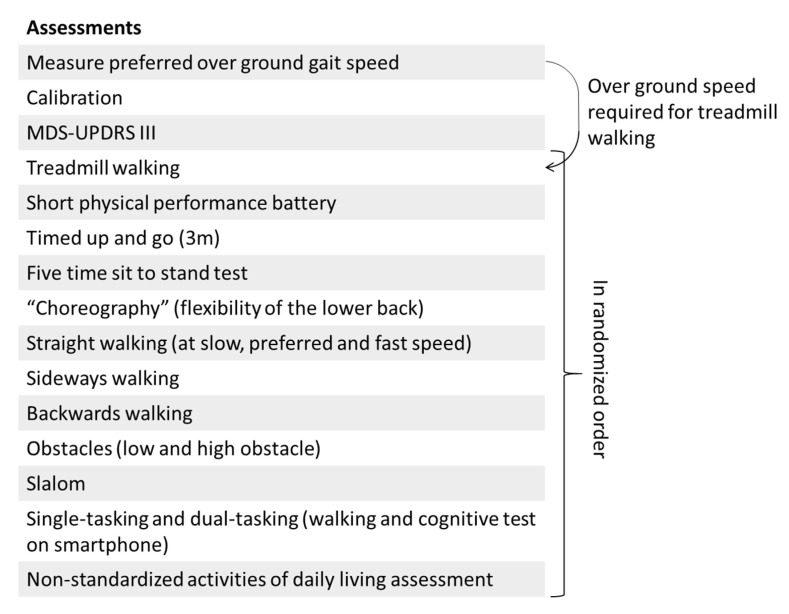
Overview of the protocol. The first three assessments will be performed in this fixed order, the remaining assessments will be performed in randomized order. An explanation of each assessment is provided in the text.

**Figure 3 sensors-21-05833-f003:**
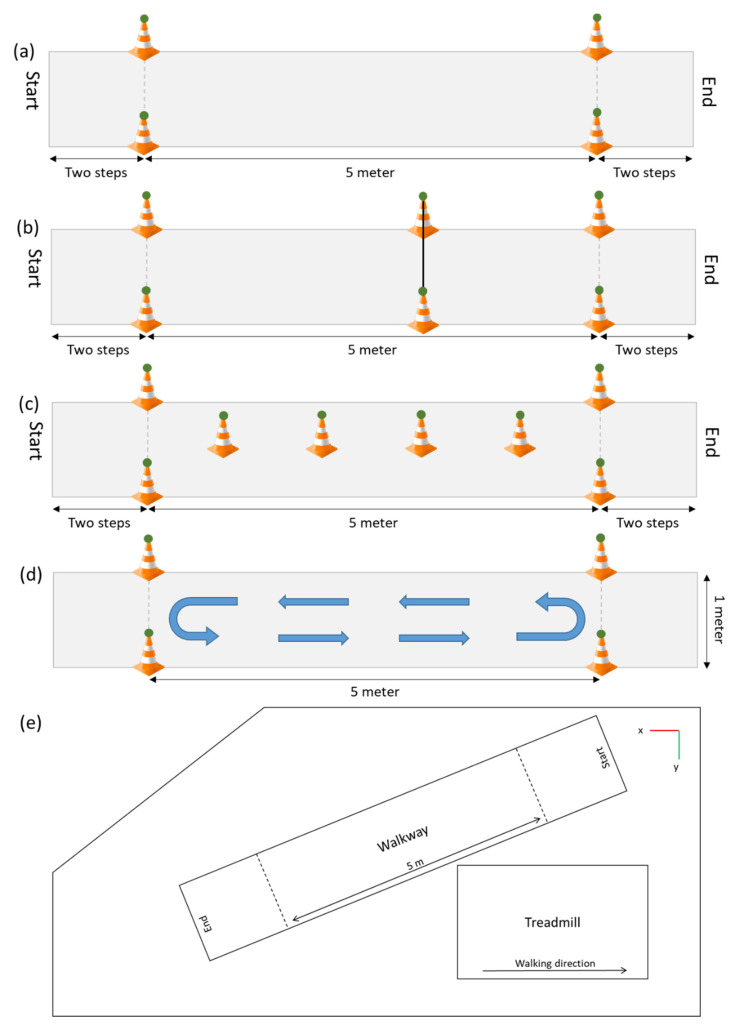
The walkway for the different over ground walking trials. (**a**) The walkway without any extra attributes, (**b**) The walkway with the obstacle, (**c**) The walkway for the slalom assessment, (**d**) The walkway for the dual-tasking assessments. The green circles represent the reflective markers captured with the optical motion capture system. (**e**) Top view of the laboratory with the orientation of the optical motion capture system (right-handed coordinate system).

**Figure 4 sensors-21-05833-f004:**
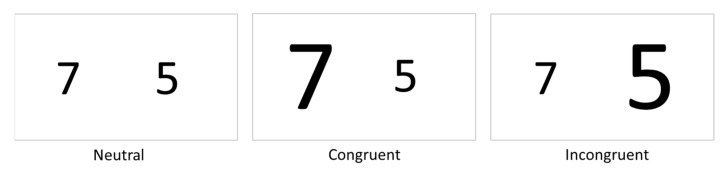
The three conditions of the numerical Stroop test that will be performed on smartphone.

## Data Availability

Data of the first 10 participants are available online (https://github.com/neurogeriatricskiel/Validation-dataset, accessed on 24 August 2021). More data can be made available upon request during the study.

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
