# Peer review of "Proposed Mobility Assessments with Simultaneous Full-Body Inertial Measurement Units and Optical Motion Capture in Healthy Adults and Neurological Patients for Future Validation Studies: Study Protocol"

_sensors, 2021, doi:10.3390/s21175833_

Round 1

Reviewer 1 Report

The manuscript describes an ambitious study to collect movement data using synchronous measurements with 15 inertial measurement units and an optical, marker-based motion capture system. A number of different movement tasks are specified.

The manuscript is well-written, reasonably detailed in the descriptions, and makes good use of figures to illustrate the planned study. Overall, the methodology is sound, but there are a few aspects that are lacking or should be improved:

1) Safety of the participants. The study includes older- and patient-populations. There is no description of how the safety of the participants will be ensured.

2) The dual-task condition. The authors describe visual cognitive tasks involving a tablet, both a reaction test and stroop tests. What is missing is a good motivation for choosing these particular dual task conditions, with relevant references to published studies. As a minor detail, it seems not to make sense to in one trial only using the incongruent mode, since the user may pick up on this fact and easily detect the small (in font size) number as the larger one (in value). Trials with random congruent/incongruent size/value makes more sense.

 3) Confidentiality and data security. The study protocol should specify how the various types of data collected will be securely stored and shared among researchers, and how the confidentiality/anonymity  of the data is ensured.  

Author Response

The manuscript describes an ambitious study to collect movement data using synchronous measurements with 15 inertial measurement units and an optical, marker-based motion capture system. A number of different movement tasks are specified.

The manuscript is well-written, reasonably detailed in the descriptions, and makes good use of figures to illustrate the planned study. Overall, the methodology is sound, but there are a few aspects that are lacking or should be improved:

Comment 1: Safety of the participants. The study includes older- and patient-populations. There is no description of how the safety of the participants will be ensured.

Response 1: We thank the reviewer for reviewing and evaluating the manuscript. We agree with the reviewer that the description of safety measures is lacking, therefore the following sentences were added:

All participants will wear a safety harness that is suspended from the ceiling while walking at the treadmill. (line 182)

During the assessment of patients, there will always be a staff member standing close to the participant to support the patient in case of a loss of balance. (line 157)

Comment 2: The dual-task condition. The authors describe visual cognitive tasks involving a tablet, both a reaction test and stroop tests. What is missing is a good motivation for choosing these particular dual task conditions, with relevant references to published studies. As a minor detail, it seems not to make sense to in one trial only using the incongruent mode, since the user may pick up on this fact and easily detect the small (in font size) number as the larger one (in value). Trials with random congruent/incongruent size/value makes more sense.

Response 2: We thank the reviewer for this constructive feedback. We added an explanation why we have these two tasks in the protocol and why we have two dual-tasks with different complexity. Each numerical Stroop test contains 8 responses of each condition (neutral, congruent and incongruent), so 24 responses in total. The order of the 24 options is randomized as explained in the “Single and dual-tasking” section.

The following text was added to the methods section:

It is know that the complexity of the dual-task influences the dual-task costs [31,32], therefore two tasks with different complexity will be performed. (line 238)

A more complex reaction time test that is more often used to measure the dual-task performance is the Stroop test [33–35]. The Stroop test also measures the cognitive inhibition [35,36]. In this study a numerical Stroop test will be performed. (line 246)

Comment 3: Confidentiality and data security. The study protocol should specify how the various types of data collected will be securely stored and shared among researchers, and how the confidentiality/anonymity  of the data is ensured.

Response 3: The following paragraph was added to the methods section to address these comments:

2.7          Database and data availability

The demographics and clinical data will be stored in a research electronic data capture (REDCap) database hosted at Kiel University [37]. This data will be shared upon reasonable request.

The IMU and optical data will be stored on a server of Kiel University that is only accessible by the research team. These data will also be made available online. The data of the first five healthy young adults and five older adults that have been measured already are available as “.mat” files in an open data repository (https://github.com/neurogeriatricskiel/Validation-dataset). More information about the data files can be found in the supplementary material.

The videos that will be recorded will be stored on a separate server of the University Hospital of Kiel and will only be accessible to a small part of the research team. The videos will not be shared since they contain identifying information. (Page 9)

Reviewer 2 Report

This paper sought to define a protocol based on the integrated use of both inertial measurement units (IMUs) and optoelectronic marker-based system addressing the possibility to implement a (large) dataset including the measurements – performed on healthy adults, healthy older adults (age > 60) and neurological patients (e.g., Parkinson’s disease before and after ) -, of both standardized mobility tasks (i.e., ground straight/sideways/backwards walking, obstacle, slalom, treadmill walking, reaction time during walking, TUG, SPPB, choreography, etc.) and non-standardized daily-life activities (i.e., setting a table, tooth brushing, sitting and talking, etc.); the authors aimed to collect important clinical information by using also specific scales and questionnaires (e.g., Charlson Comorbidity Index, Montreal Cognitive Assessment, EQ-5D-5L, VAS, Expanded Disability Status Scale, etc.). Eventually, the defined dataset would be the basis for developing and validating full-body IMU-based solutions, approaches and algorithms, able to quantify the mobility in both healthy adults and neurological patients.    

General comment

This paper sought to define the basis to provide a large mobility dataset containing motion data of both healthy adults and neurologic patients performing both standardized tasks and non-standardized activities mimicking the daily-life activities. The protocol seems to be indeed complete and coherent with this main objective. However, there are several issues to solve before considering publication. First of all, this paper can not be considered an original research article, since it does not contain any real acquired data; therefore, I would suggest the authors to change the type of submission from “article” to “communication”. Furthermore, the paper – as it is – does not contain any novelty, since the dataset is not yet defined; indeed, the paper can be considered as a structured description of the methodologies that are going to be used in a future study. On the other hand, if the perspective of the studio were a little broader, addressing the possibility to define a “harmonized” protocol that can be shared among different centers, the work would acquire value.

The structure of the article is in general correct, including Introduction, Materials and Methods [with subheading], Discussion and Conclusions. There is no Results section, due to the inherent nature of the paper.

The protocol seems to be correctly reported but further details are needed; several concerns are here reported to the authors.

The use of the English language seems to be correct for a non-native English speaker. The paper is well written and presented.

Title

If you agree with my suggestion, I would stress better in the title that this paper is a proposal of a possible harmonized protocol.

Abstract

Ok. In general, well written. I would only underline that this paper does not provide any dataset, but – on the other hand – proposes a complete and coherent protocol that can be used to define a possible dataset, in the near future.

Introduction

In general, ok. It is well written and report all the information required to understand the need for a protocol and for the definition of a dataset. Even in this section, I would underline better that the aim of this paper is not to provide a dataset, but to define a protocol that can be shared to build a potentially very large dataset.

Materials and Methods

In general, also this section is well written. Just few notes:

  • Line 75-79: ok reporting the ethical approval, but indeed here you did not apply the protocol on any subjects or patients. Further, I would see all the paper in a wider perspective, able to include also other clinical centers.
  • Line 85-88: even concerning recruitment, I would consider the possibility to extend the protocol to further clinical centers. This description is more coherent with a defined study applied on a well-defined cohort of subjects and patients and presenting data about it.
  • Line 115 – 132: Within the very same perspective of providing information to develop a coherent acquisition protocol, I would not underline the use of a specific system or solution; it would be better to define the general technical requirements (number of sensors, number of cameras, location of sensors and markers, sampling rate, volume size, resolutions, accuracy, precision, full-scale, types of synchronization, external cameras, etc.). It is fundamental to also define the calibration procedure for both the systems and, above all, how you define the anatomical frame references with respect to the technical ones; if you borrowed this information from literature, please report in detail the articles.
  • Line 121: Why an additional IMU in the pocket? Please justify this choice.
  • Line 119: Why a distal sensor on the “ankle”?
  • I would also add a section that can describe the format you required to share the data, and – if possible – how the dataset is technically defined and if it can be placed on an open repository (it would be great!).

If the general “sharing” perspective is not aligned with your idea of paper, anyhow start from a general approach that then can be declined with respect to the instrumentation and facilities you have.

Discussion

Ok. Well written.

Conclusion

Ok. Well written.

References

Literature is extended and up-to-date.

Figures

Ok.

Author Response

This paper sought to define a protocol based on the integrated use of both inertial measurement units (IMUs) and optoelectronic marker-based system addressing the possibility to implement a (large) dataset including the measurements – performed on healthy adults, healthy older adults (age > 60) and neurological patients (e.g., Parkinson’s disease before and after ) -, of both standardized mobility tasks (i.e., ground straight/sideways/backwards walking, obstacle, slalom, treadmill walking, reaction time during walking, TUG, SPPB, choreography, etc.) and non-standardized daily-life activities (i.e., setting a table, tooth brushing, sitting and talking, etc.); the authors aimed to collect important clinical information by using also specific scales and questionnaires (e.g., Charlson Comorbidity Index, Montreal Cognitive Assessment, EQ-5D-5L, VAS, Expanded Disability Status Scale, etc.). Eventually, the defined dataset would be the basis for developing and validating full-body IMU-based solutions, approaches and algorithms, able to quantify the mobility in both healthy adults and neurological patients.    

General comment

This paper sought to define the basis to provide a large mobility dataset containing motion data of both healthy adults and neurologic patients performing both standardized tasks and non-standardized activities mimicking the daily-life activities. The protocol seems to be indeed complete and coherent with this main objective. However, there are several issues to solve before considering publication. First of all, this paper can not be considered an original research article, since it does not contain any real acquired data; therefore, I would suggest the authors to change the type of submission from “article” to “communication”. Furthermore, the paper – as it is – does not contain any novelty, since the dataset is not yet defined; indeed, the paper can be considered as a structured description of the methodologies that are going to be used in a future study. On the other hand, if the perspective of the studio were a little broader, addressing the possibility to define a “harmonized” protocol that can be shared among different centers, the work would acquire value.

The structure of the article is in general correct, including Introduction, Materials and Methods [with subheading], Discussion and Conclusions. There is no Results section, due to the inherent nature of the paper.

The protocol seems to be correctly reported but further details are needed; several concerns are here reported to the authors.

The use of the English language seems to be correct for a non-native English speaker. The paper is well written and presented.

 Response: We thank the reviewer for reviewing and evaluating the manuscript. Regarding the type of submission, we submitted it as “article” since we found another study protocol paper in Sensors that was submitted as “article”. We are however also fine with changing it to “communication”. We will leave this decision to the editor.

As partners of the IMI Mobilise-D consortium, we understand the difficulty of harmonizing data collection and analyses pipelines across multiple sites, with less equipment and protocols. The measurement setup of this project is very extensive (47 reflective markers, 16 IMUs, about 30 trials and ~3h of measuring) and not a lot of centers could warrant the expected data quality. However, if the data quality can be assured and the data collection procedures are within reasonable deviations we are happy to collaborate and include data from other research centers. The idea of this particular data set is to provide a world unique data set allowing researchers from all over the world to develop and improve current 3D motion capture and IMU algorithms during lab assessment. In the meantime, data of already 10 subjects have been collected and processed. The data is uploaded and ready to use for the research community, the link to the data has been added to the manuscript. The simulated home like environment assessment is the right step into a real world gait and mobility assessment as the transfer to the real world is realistic. If you have suggestions or laboratories in mind, please do not hesitate to contact us directly. We will make the protocol and the data publicly available and share it on our media outlets e.g. Twitter and neurogeriatrics-kiel.de. 

Title

If you agree with my suggestion, I would stress better in the title that this paper is a proposal of a possible harmonized protocol.

Response: We agree with the suggestion of the reviewer and changed the title to: “Proposed mobility assessments with simultaneous full-body inertial measurement units and optical motion capture in healthy adults and neurological patients for future validation studies: Study protocol”

Abstract

Ok. In general, well written. I would only underline that this paper does not provide any dataset, but – on the other hand – proposes a complete and coherent protocol that can be used to define a possible dataset, in the near future.

Response: We changed the following sentence to underline that we propose here a study protocol:

This study proposes a protocol for a dataset that can be used to develop and validate IMU-based mobility algorithms for healthy adults (18-60 years), healthy older adults (>60 years), and patients with Parkinson’s disease, multiple sclerosis, a symptomatic stroke and chronic low back pain. (line 25)

Introduction

In general, ok. It is well written and report all the information required to understand the need for a protocol and for the definition of a dataset. Even in this section, I would underline better that the aim of this paper is not to provide a dataset, but to define a protocol that can be shared to build a potentially very large dataset.

Response: Small changes were made to the following two sentences to clarify that we still need to build the dataset (lines 64 and 70):

We propose here a study protocol to build a full-body mobility dataset of healthy young and older participants and neurological patients, including PD, MS, stroke and chronic low back pain (CLBP).

The aim of the study is to build a dataset for the research community that can be used to develop and validate IMU-based mobility algorithms for healthy adults and neurological patients.

Materials and Methods

In general, also this section is well written. Just few notes:

  • Line 75-79: ok reporting the ethical approval, but indeed here you did not apply the protocol on any subjects or patients. Further, I would see all the paper in a wider perspective, able to include also other clinical centers.
    • Response: We have recently started the data collection for this study and have therefore added the data of the first few participants to this revision. See the earlier response regarding including also other clinical centers.
  • Line 85-88: even concerning recruitment, I would consider the possibility to extend the protocol to further clinical centers. This description is more coherent with a defined study applied on a well-defined cohort of subjects and patients and presenting data about it.
    • Response: Since we have not found any other centers that can and want to participate in this study, it is indeed described as a defined study that will be applied to a defined cohort at probably only one center.
  • Line 115 – 132: Within the very same perspective of providing information to develop a coherent acquisition protocol, I would not underline the use of a specific system or solution; it would be better to define the general technical requirements (number of sensors, number of cameras, location of sensors and markers, sampling rate, volume size, resolutions, accuracy, precision, full-scale, types of synchronization, external cameras, etc.). It is fundamental to also define the calibration procedure for both the systems and, above all, how you define the anatomical frame references with respect to the technical ones; if you borrowed this information from literature, please report in detail the articles.
    • Response: We thank the reviewer for these suggestions. We prefer to keep the information about the specific equipment in the manuscript. Since we recently started the data collection, we have added a link to this data to the manuscript. More data can be made available on request during the study. For the researchers that will work with this data it will be useful to know certain specifications about the equipment that will be used. Moreover, future studies that will analyse the data that will be collected with this study, can just refer back to this manuscript for all the details and will not have to describe several aspects again.

Regarding the calibration procedure, the following sentence was added to provide more information about how the calibration trial can be used (line 175):

This calibration trial can be used to define the anatomical reference frame with respect to the technical reference frame for both the IMU and the optical motion capture system [28–30].

  • Line 121: Why an additional IMU in the pocket? Please justify this choice.
    • Response: The additional IMU in the pocket could be used to validate algorithms that were developed for smartphones that participants wear in the pocket (e.g. 10.1016/j.gaitpost.2017.09.030; 10.1089/tmj.2017.0215). The following sentence was added to the manuscript (line 122):

The data from this 16th IMU could be used to develop and validate algorithms for smartphones that are commonly worn in the pocket.

  • Line 119: Why a distal sensor on the “ankle”?
    • Response: An IMU positioned just above the ankle joint is a position that we use in multiple studies in our research group, but is also used more frequently in other studies (e.g. 10.3390/s20102858; 10.1016/j.gaitpost.2017.04.013)
  • I would also add a section that can describe the format you required to share the data, and – if possible – how the dataset is technically defined and if it can be placed on an open repository (it would be great!).
    • Response: We thank the reviewer for the constructive feedback. We have decided to add an additional subsection to the methods section that provides information about the dataset and the availability of the dataset. The following text was added to the manuscript:

2.7           Database and data availability

The demographics and clinical data will be stored in a research electronic data capture (REDCap) database hosted at Kiel University [37]. This data will be shared upon reasonable request.

The IMU and optical data will be stored on a server of Kiel University that is only ac-cessible by the research team. These data will also be made available online. The data of the first five healthy young adults and five older adults that have been measured already are available as “.mat” files in an open data repository (https://github.com/neurogeriatricskiel/Validation-dataset). More information about the data files can be found in the supplementary material.

The videos that will be recorded will be stored on a separate server of the University Hospital of Kiel and will only be accessible to a small part of the research team. The videos will not be shared since they contain identifying information. (Page 9)

If the general “sharing” perspective is not aligned with your idea of paper, anyhow start from a general approach that then can be declined with respect to the instrumentation and facilities you have.

Discussion

Ok. Well written.

Conclusion

Ok. Well written.

References

Literature is extended and up-to-date.

Figures

Ok.

Round 2

Reviewer 2 Report

This paper aimed to define a protocol based on the integrated use of both inertial measurement units (IMUs) and optoelectronic marker-based system addressing the implementation of a large dataset including the measurements – performed on healthy adults, healthy older adults (age > 60) and neurological patients (e.g., Parkinson’s disease before and after ) -, of both standardized mobility tasks (i.e., ground straight/sideways/backwards walking, obstacle, slalom, treadmill walking, reaction time during walking, TUG, SPPB, choreography, etc.) and non-standardized daily-life activities (i.e., setting a table, tooth brushing, sitting and talking, etc.). The authors aimed to collect important clinical information by using also specific scales and questionnaires (e.g., Charlson Comorbidity Index, Montreal Cognitive Assessment, EQ-5D-5L, VAS, Expanded Disability Status Scale, etc.). Eventually, the defined dataset will be the basis for developing and validating full-body IMU-based solutions, approaches and algorithms, able to quantify the mobility in both healthy adults and neurological patients. The authors did answer and/or justified all the comments and issues arisen during the first round of review. The perspective given by the authors within the paper is indeed interesting, although different from what I initially understood.